# Fetal Red Blood Cells: A Comprehensive Review of Biological Properties and Implications for Neonatal Transfusion

**DOI:** 10.3390/cells13221843

**Published:** 2024-11-07

**Authors:** Claudio Pellegrino, Elizabeth F. Stone, Caterina Giovanna Valentini, Luciana Teofili

**Affiliations:** 1Dipartimento di Scienze di Laboratorio ed Ematologiche, Fondazione Policlinico Universitario “A. Gemelli” IRCCS, 00168 Rome, Italy; claudio.pellegrino01@icatt.it (C.P.); caterinagiovanna.valentini@policlinicogemelli.it (C.G.V.); 2Sezione di Ematologia, Dipartimento di Scienze Radiologiche ed Ematologiche, Università Cattolica del Sacro Cuore, 00168 Rome, Italy; 3Department of Pathology & Cell Biology, Columbia University Irving Medical Center, New York, NY 10032, USA; es2024@cumc.columbia.edu

**Keywords:** red blood cell transfusion, fetal hemoglobin, cord blood, preterm neonates

## Abstract

Transfusion guidelines worldwide include recommendations regarding the storage length, irradiation, or even donor cytomegalovirus serostatus of red blood cell (RBC) units for anemic neonates. Nevertheless, it is totally overlooked that RBCs of these patients fundamentally differ from those of older children and adults. These differences vary from size, shape, hemoglobin composition, and oxygen transport to membrane characteristics, cellular metabolism, and lifespan. Due to these profound dissimilarities, repeated transfusions of adult RBCs in neonates deeply modify the physiology of circulating RBC populations. Unsurprisingly, the number of RBC transfusions in preterm neonates, particularly if born before 28 weeks of gestation, predicts morbidity and mortality. This review provides a comprehensive description of the biological properties of fetal, cord blood, and neonatal RBCs, including the implications that neonatal RBCs, and their replacement by adult RBCs, may have for perinatal disease pathophysiology.

## 1. Introduction

Fetal and neonatal red blood cells (RBCs) fundamentally differ from those of older children and adults. They vary in size, shape, hemoglobin composition, oxygen transport, membrane characteristics, cellular metabolism, and lifespan. Due to these profound differences, repeated transfusions of adult RBCs in neonates, particularly if born extremely premature, deeply modify the physiology of circulating RBC populations. This review provides a comprehensive description of the biological properties of fetal, cord blood, and neonatal RBCs, including the implications that neonatal RBCs, and their replacement by adult RBCs, may have for perinatal disease pathophysiology. To this end, we define each cell type as follows: “fetal RBCs” circulate in the fetus and contain only or mainly hemoglobin F (HbF); “cord blood RBCs” (CB-RBCs) are collected from the umbilical cord immediately after delivery; and “neonatal RBCs” are collected from neonates after delivery, either by phlebotomy or capillary blood sampling.

## 2. Fetal RBC Shape, Hemoglobin Content, and Morphology

Modern hematology analyzers produce large laboratory datasets that allow the definition of reference ranges for RBC indices in neonates born at different gestational ages [1,2,3]. Between 25 and 40 weeks of gestation, the median mean corpuscular volume (MCV) decreases from 119 fl to 106 ± 4 fl, while the mean corpuscular hemoglobin (MCH) declines from 40 pg to 36 pg. Consequently, the mean corpuscular hemoglobin concentration does not change appreciably with gestational age (median: 34 pg/dL) [3]. Overall, hemoglobin levels and hematocrit increase throughout gestation, and fetal RBCs remain larger, with a higher hemoglobin content, than adult RBCs [1]. In addition, delayed cord clamping, which is considered the standard of care in term delivery [4] and may reduce the risk of death before discharge for preterm neonates [5], also influences neonatal hemoglobin and hematocrit. For example, in term neonates the mean hemoglobin and hematocrit 24–48 h after delivery with immediate cord clamping were 17.8 g/dL and 53%, respectively, while after delayed cord clamping they increased to 19.4 g/dL and 58.4%, respectively, without inducing hyperviscosity or bilirubinemia [6]. Additionally, in preterm neonates, hemoglobin and hematocrit significantly increased after delayed cord clamping (18.7 g/dL and 53.9%, respectively), as compared to immediate cord clamping (16.8 g/dL and 48.8%, respectively); these persisted for ~7 days after birth along with a significant increase in phototherapy for hyperbilirubinemia in the delayed cord clamping cohort [7].

Neonatal RBC reference ranges do not convey the intrinsic heterogeneity of these RBCs. For example, red blood cell distribution width (RDW) reference intervals are higher for neonates than for older children and adults; this is even more pronounced in preterm infants due to anisocytosis [8]. Indeed, neonates have markedly higher reticulocyte levels as compared with adults [9], and term and preterm newborn RBCs display at conventional light microscopy more morphological anomalies as compared with adults, complicating the diagnosis of RBC disorders at birth. Thus, only ~40% of term and preterm fetal RBCs were classical biconcave discs, whereas up to 20% had altered morphology, including echinocytes, acanthocytes, dacrocytes, and schistocytes [10]. By micropipette aspiration, neonatal RBCs had a larger volume, surface area, and diameter as compared with adult RBCs, resulting in a decreased surface-area-to-volume ratio [11]. In addition, in 62 preterm infants of gestational age 22 + 0–26 + 6, 1.5–24.9% of RBCs at birth had a particularly high hemoglobin content (i.e., Hyper-He; >49 pg/RBC) as compared with <1% Hyper-He in adults, perhaps from stress erythropoiesis or the persistence of immature erythromyeloid progenitors [12].

Quantitative phase imaging of cord RBCs [13] revealed that RBCs of term newborns had a significantly different morphology from those of nonpregnant women: the volume and surface area of cord RBCs were larger than RBCs from the nonpregnant women, Hb content was significantly higher in cord RBCs and newborn RBCs as compared with RBCs from nonpregnant adults and maternal RBCs, and cord RBCs exhibited less sphericity as compared with those from nonpregnant women. Notably, RBCs with low sphericity (e.g., discocytes) benefit when passing through narrow capillaries [14]; nonetheless, membrane fluctuations, a proxy for cellular deformability, of cord RBCs were not significantly altered as compared to adult RBCs [13].

Table 1 summarizes changes of hematological indices and morphological parameters of fetal RBCs in comparison with adult RBCs, and the study references providing this evidence.

## 3. Membrane Properties

RBC membranes comprise a lipid bilayer containing integral membrane proteins attached to the underlying cytoskeleton; these, together with intracellular viscosity, allow RBC deformation to handle shear stress and perfuse narrow capillaries [15,16] (see the subsection on rheology below). Key differences between fetal, neonatal, and adult RBC membranes include lipid content [17,18,19,20], membrane fluidity [21,22,23], and membrane protein concentrations [24,25]; nonetheless, it is not clear whether fetal/neonatal RBCs differ from adult RBCs regarding osmotic fragility [26] (see below).

Lipid and fatty acid compositions of cord and neonatal RBCs differ from adult RBCs. Although the total lipid content per cell is greater in cord than adult RBCs, and although the relative compositions of phospholipid and cholesterol were similar in one classic study [17], the cholesterol: phospholipid ratio in CB-RBCs was significantly greater than in RBCs from normotensive pregnant women in another study [18]. Small alterations in phospholipid and fatty acid content in CB-RBCs were reported [17], whereas other groups reported increased concentrations of polyunsaturated fatty acids (PUFAs) directly proportional to gestational age [19,20]. Arachidonic acid (ARA, ω-6 PUFA, 20:4) and docosahexaenoic acid (DHA, ω-3 PUFA, 22:6) are PUFAs critically important for retina and brain development. The active placental transfer of PUFAs and other fatty acids from maternal stores is important for the fetus, as the fatty acid composition of cord blood reflects that of maternal blood [27]. This transfer is highly dynamic, considering that biomagnification (i.e., the fetus receives higher circulating levels than the mother) and bioattenuation (i.e., fetal levels become lower than those of the mother) were described. For example, during the third trimester, the placenta preferentially transfers ARA and DHA from mother to fetus [28]. DHA and ARA levels were reduced in RBC membranes of infants with retinopathy of prematurity (ROP, a visual impairment frequent in extreme preterm neonates) as compared with premature infants who did not develop ROP [29], and enteral lipid supplementation with AA and DHA may reduce the risk of severe ROP in extremely preterm infants [30]. Moreover, higher PUFA levels in the perinatal period positively impact neurodevelopmental outcomes in school-age children [31].

In addition to dietary factors, enzyme activity levels and genetic polymorphisms may contribute to fetal RBC membrane lipid composition. For example, fatty acid desaturases (FADs) in adult RBCs assist in regulating PUFA levels and NADH homeostasis; therefore, they are relevant for donor RBC adaptation to refrigerated storage and post-transfusion recovery [32]. When lipid ratios were measured as a proxy for FAD indices over the first 10 weeks of life, FAD activity levels were altered when comparing preterm to term infants, although the rates of change over time were similar [33]. In addition, single nucleotide polymorphisms (SNPs) in the FADS1 gene of pregnant women and their infants can alter the availability of long-chain PUFAs [34]; however, these results may not necessarily translate to long-term PUFA levels, because DHA supplementation during gestation indicated that dietary PUFA content was more important for metabolic outcomes in older children [35].

Lipid composition and integral proteins influence RBC membrane fluidity. When measured by hydrophobic 1,6-diphenyl-1,3,5,-hexatriene (DPH) fluorescence anisotropy, membrane fluidity was significantly greater in human newborns’ RBCs as compared with those from adults, indicating a lower degree of fluidity in the internal hydrophobic core [21,22]. In contrast, using polar probes, the freedom of motion in the hydrophilic lipid headgroup regions of the membranes was similar for both cell types [22]. The differences observed were attributed to differences in lipid composition, vitamin E levels, and increased lipid oxidation [22]. Nonetheless, fetal RBC membrane fluidity seems not to be influenced by gestational age [23].

Given the differences in membrane fluidity when comparing neonatal and adult RBCs and given that integral membrane proteins are anchored to the cytoskeleton and affect RBC deformability, the membrane protein compositions of fetal, neonatal, and adult RBCs were characterized. When assessed by gel electrophoresis [24], RBC membrane protein patterns from premature infants, term infants, and adults were not qualitatively different. In addition, a comprehensive proteomic analysis of cord and adult RBCs identified the expected differences in hemoglobin components; in addition, five adult RBC proteins had higher expression, and three cord RBC proteins had higher expression [25] (Figure 1(1)–(5)). Thus, carbonic anhydrase 1 and 2 and aquaporin 1, all linked to gas and bicarbonate–chloride exchange [36], and BCAM and Semaphorin 7A, which carry the Lutheran and JMH blood group antigens, respectively, had higher expression in adult RBCs. In contrast, three myosin subunits that interact with actin (i.e., myosin heavy chain 9 and 10, and myosin regulatory light chain 12A), had higher expression in cord RBCs. These findings may partially explain the membrane deformability differences seen between cord and adult RBCs [25] (Figure 1(2)).

From an immunological perspective, the RBC membrane of newborns also differs in antigen expression. For example, blood group I antigen is highly expressed on fetal and neonatal RBCs, gradually decreasing in a reciprocal relationship with the I antigen during early postnatal life (Figure 1(1)); in addition, Lewis blood group antigens are poorly expressed on RBCs at birth [37].

Cord RBCs have a larger osmotically inactive fraction as compared with adult RBCs. The hydraulic conductivity and osmotic permeability to glycerol of cord RBCs differed as compared with adult RBCs [38]. In addition, fetal RBCs exhibit lower channel-mediated water permeability [39], higher passive transport of potassium ions through the K+/Cl− cotransporter channel [40], and higher levels and activity of Na+/K+-ATPase [41]. RBC osmotic fragility relates to the ability of their membranes to maintain structural integrity when exposed to osmotic stress. Reports vary regarding the relative osmotic fragility of RBCs in newborns and adults in relation to the different experimental conditions. With a hypotonic saline and glycerol lysis test, RBCs of full-term and, to a greater extent, preterm neonates appear more resistant to osmotic hemolysis than adult RBCs [26]. However, cord RBCs are more heterogeneous than adult RBCs and hemolyze over a wider range of hypotonic concentrations, suggesting the presence of subpopulations with differential osmotic sensitivity [26]. When exposed to the more extreme conditions required for cryopreservation using high glycerol or saline concentrations, cord RBCs were more sensitive to shrinkage and swelling, indicating the need to optimize cryopreservation protocols for cord RBCs [42].

## 4. Rheology

RBCs must deform sufficiently to perfuse small blood vessels. This is mainly determined by RBC shape and surface area, dynamic cytoskeletal changes during shear stress, cytosolic viscosity, and membrane resistance to extension and bending forces [15,16]. Static rigidities (defined by elastic moduli for extension and bending) limit RBCs’ ability to enter small channels at given pressures, whereas dynamic rigidities (characterized by time constants for elastic recovery from extension and bending deformations) influence the rate of entry [16].

The rheologic properties of neonatal RBCs subtly differ from adult RBCs [43]. For example, neonatal RBC deformability and viscoelastic properties deviate only slightly from those of adult RBCs; specifically, membrane surface viscosity and hemoglobin viscosity were similar for both cell types, whereas neonatal RBCs had lower resistance to extensional and folding deformation and higher time constants for shape recovery [16]. Although one subsequent study found no significant differences in deformability among RBCs from fetuses, preterm and term neonates, and adults at any shear stress [44], subtle differences in mechanical properties between adult and fetal RBCs were detected by flow-based deformation cytometry and deep neural network analysis [45]. Additionally, the aspiration pressure of neonatal RBCs into micropipettes is increased [16] and filterability through small pores is reduced [22,23,46]; these differences were associated exclusively with the larger size of these RBCs, without significant influences from differential membrane lipid composition. Although the larger size of fetal and neonatal RBCs would be expected to impair flow in narrow vessels with diameters below 3.6–4.1 μm, this appears to be ameliorated by lower plasma viscosity in vivo in fetuses and neonates [47].

Mechanical stress deforms RBC membranes until a yield point is reached, beyond which additional forces cause irreversible membrane deformation and RBC rupture. Neonates have a subpopulation of extremely rigid and dense RBCs, which may result from decreased surface area due to membrane fragmentation and increased cellular viscosity [48]. Using micropipette techniques and a flow channel system, the calculated critical shear force to induce membrane failure was similar for neonatal and adult RBCs, but the rate of membrane failure was higher for neonatal RBCs, suggesting that they more rapidly lose membrane once the yield shear force is exceeded [49]. In addition, preterm and term neonatal RBCs, as compared with adult RBCs, form tethers at lower shear stress and deforming force, potentially explaining their increased anisocytosis and decreased lifespan [50].

RBCs form rouleaux during stasis or under low shear stress [51]. Rouleaux formation is markedly decreased in small preterm infants at birth as compared with term neonates and adults; this may partially be due to decreased plasma viscosity and differing plasma composition, as these results were not seen in cross-suspension studies (e.g., when neonatal RBCs were suspended in adult plasma or vice versa), or when neonatal and adult RBCs were resuspended in a dextran solution [52,53]. However, RBC-specific factors may also be relevant, in that newborn RBCs exhibit reduced rouleaux formation and higher rouleaux disaggregation as compared with maternal RBCs; in addition, resuspending neonatal RBCs in maternal plasma or maternal RBCs in neonatal plasma did not fully reverse this phenotype, suggesting that adult and neonatal RBCs themselves contribute to rouleaux formation [54]. Moreover, when RBCs from preterm neonates were suspended in neonatal plasma there was minimal or no rouleaux formation as compared with adult RBCs; however, when RBCs from preterm neonates were suspended in a dextran solution the resulting rouleaux had enhanced resistance to disaggregation by flow as compared with adult RBC rouleaux, potentially due to stronger cell–cell interactions of neonatal RBCs [55].

These studies were conducted in closed systems in artificially created environments; therefore, they represent a simplification of the situation in vivo which involves dynamic interactions between endothelium, plasma, RBCs, white blood cells, and platelets [56], thereby limiting the generalizability of these findings. Moreover, RBC deformability is important for transfusion outcomes, correlating with hemodynamic functioning of circulating RBCs, hemolysis in vivo, and tissue oxygenation [57]. For example, the deformability of RBCs from adult blood donors is reduced as compared with RBCs from preterm neonates; in contrast, the deformability of autologous CB-RBCs is comparable to preterm neonatal RBCs [58]. Thus, the uniqueness of neonatal RBCs and the neonatal microcirculation suggest that transfusing neonates with adult RBCs may have significant rheological effects.

## 5. Hemoglobin

The most obvious difference between adult and fetal RBCs is their expression of hemoglobin chains. Human hemoglobin molecules are closely related, formed by symmetric pairing of dimer polypeptide chains into a tetrameric functional structure: fetal hemoglobin (HbF) consists of α2γ2 tetramers, while the major form of adult hemoglobin (HbA) consists of α2β2 tetramers [59].

After approximately 8 weeks of gestation, HbF synthesis replaces embryonic hemoglobin (HbE) and predominates during gestation; around parturition a progressive downregulation of γ chains is paralleled by increased synthesis of β chains [60]. The regulation of HbF synthesis is an active field of research due to the benefits of increased HbF levels in patients with sickle cell disease and β-thalassemia [61]. The chief driver of this switch is the BCL11A transcriptional repressor, which occupies γ globin gene promoters, inhibiting their activity in adult erythroblasts (Figure 1(7)) [62]. The HbF to HbA switch is not complete or irreversible, and some erythroid precursors that can synthesize HbF persist in adult life.

The physiological switch from HbF to HbA synthesis closely correlates with the fetal developmental maturation stage and is not perturbed by premature exposure to the extrauterine environment. Thus, HbF levels in the first weeks of life in preterm infants born at 27–32 weeks of gestation did not differ dramatically from those of fetuses at the same development stage [63]. The slow transition toward HbA synthesis accelerated as the preterm infant approached 38 weeks post-conception; indeed, there were no statistical differences between HbA levels in preterm newborns who had spent several weeks or months of extrauterine life and term newborns [63].

Hemoglobin’s major role is oxygen transport, which is finely tuned by allosteric regulation intrinsic to the Hb molecule, or by extrinsic interactions with other molecules, such as 2,3-diphosphoglycerate (2,3-DPG), anions, and protons. The association between Hb and 2,3-DPG increases oxygen offloading from Hb to tissues; conversely, reduced interactions with 2,3-DPG induce Hb to bind oxygen more tightly. HbF γ chains have a lower binding capacity for 2,3-DPG than HbA β chains; thereby as a result, HbF has a higher affinity for oxygen as compared with maternal HbA, which allows adequate oxygen delivery from mother to fetus until birth [64,65,66].

Hemoglobin in circulating RBCs rapidly reacts with nitric oxide (NO), a regulator of blood pressure and vascular homeostasis, forming methemoglobin and nitrate [67]. RBCs can produce NO via endothelial nitric oxide synthase (eNOS) and release it in both hypoxic and normoxic conditions [68]. By carrying NO, Hb helps regulate microcirculatory blood flow, contributing to the RBC’s “erythrocrine function” [68,69]. S-nitrosylation of Hb (S-nitrosohemoglobin, SNO-Hb) and denitrosylation of SNO-Hb play key roles in oxygen-dependent regulation of vasodilatation [70]. Several groups speculate that higher HbF levels correlate with increased NO bioavailability. For example, ovine HbF has increased nitrite reductase activity, translating into increased NO production [71], whereas human HbF exhibits a faster rate of oxidative denitrosylation and nitric oxide release [72]. Supporting this hypothesis, SNO-HbF levels were higher in preterm neonates at gestational age < 30 weeks as compared with those ≥30 weeks [73]. In addition, NO promotes lung maturation [74], and the early rapid postnatal decline in HbF levels is associated with the development of bronchopulmonary dysplasia (BPD) in very preterm infants [75]. Therefore, inhalated NO is frequently administered to preterm neonates to reduce pulmonary hypertension and prevent BPD [76]. Thus, fetal RBCs are active in modulating blood flow, finely tuning tissue oxygen delivery due to their increased expression/activation of eNOS under hypoxic conditions (Figure 1(8)) [77].

As compared with HbA, HbF has several additional properties relevant for RBC biology. For example, HbF has greater structural stability with better tetrameric integrity as compared with HbA, likely leading to a slower rate of intrinsic heme loss and heme-derived free iron release [78]. Additionally, free HbF induces less damage to human umbilical vein endothelial cells [78] and appears less prone to form protein-based free radicals or induce DNA cleavage [79]. HbF also has higher intrinsic pseudoperoxidase activity, which neutralizes reactive oxygen species (ROS) by a peroxide radical self-termination reaction that generates more stable molecules, thereby diminishing neonatal RBC oxidation (Figure 1(8)) [80]. Owing to its role in redox chemistry, hemoglobin may also serve as a “murzyme”, a redox enzyme contributing to RBC ATP synthesis using diffusible oxygen species [81]. As HbF differs in terms of docking affinity to adenosine nucleotides and phosphoglycerates, it may also contribute to metabolic differences between adult and neonatal RBCs [81].

## 6. Cellular Metabolism

Many enzymes crucial for fetal and neonatal RBC metabolism depend on RBC age. Because mature RBCs do not possess mitochondria, they cannot generate adenosine triphosphate (ATP) by oxidative phosphorylation; thus, they rely exclusively on glycolysis (Embden–Meyerhof–Parnas pathway) to generate ATP to fuel key cellular processes [82]. Given that neonates have increased reticulocyte counts compared to adults [9], the reticulocytes can confound assessments of fetal and neonatal RBC metabolism. Nevertheless, fetal and neonatal RBCs differ meta¬bolically from adult RBCs (Figure 1); this cannot be explained solely by reticulocytosis, because these findings were confirmed by comparing cord RBCs to reticulocyte-rich adult RBCs [83]. Nonetheless, no specific fetal RBC isozymes have been identified to date [83], even when analyzed by proteomics [25]. Similar results were found in CB-RBCs of extremely preterm neonates, suggesting that metabolic differences may represent a unique signature of fetal erythropoiesis and CB-RBC metabolism [84].

In RBCs, two critical pathways branch from glycolysis: the Rapoport–Luebering shunt, which generates 2,3-DPG, and the pentose phosphate pathway (PPP), which generates ribose phosphate and, importantly, reduces nicotinamide adenine dinucleotide phosphate (NADPH) (Figure 1(6)). RBC 2,3-DPG levels in CB-RBCs were higher than in adult RBCs [85], and increased during the 6 h post-partum [86], possibly reflecting an adaptation to extrauterine life.

RBC enzyme activities in the Rapoport–Luebering shunt in newborn infants were markedly increased for enolase (ENL) and phosphoglycerate kinase (PGK), but reduced for phosphofructokinase (PFK); thus, neonatal RBCs contain significantly more glucose-6-phosphate, fructose 6-phosphate, glyceraldehyde-3-phosphate, and dihydroxyacetone phosphate, and significantly less 2,3-DPG and phosphoenolpyruvate [87]. Subsequent studies confirmed this pattern of RBC enzyme activities in neonatal RBCs, with increased ENL and PGK levels, and decreased PFK levels [83,84,88,89,90,91]. The rate-limiting enzyme of the PPP is glucose 6-phosphate dehydrogenase (G6PD). G6PD deficiency, the most common enzymopathy worldwide, can cause hyperbilirubinemia in neonates, with potentially fatal consequences due to kernicterus and permanent neurologic sequelae. Therefore, there have been significant efforts to optimize techniques to screen neonates for G6PD deficiency and to define reference ranges for various ethnic groups [92]. G6PD activity in RBCs from neonates without G6PD deficiency is higher than that in adult RBCs [93]; in addition, there is a significantly negative correlation between gestational age and G6PD activity [94], with peak levels at 29–32 weeks of gestational age [95]. Surprisingly, in a recent case-control study, higher G6PD activity at birth was associated with the development of ROP, a condition in which oxidative stress plays a key pathogenetic role [96]. Although preliminary, this observation highlights the complexity of oxidative stress regulation in neonates, as a given molecule can serve as both an antioxidant and a pro-oxidant depending on context. Because NADPH produced by the PPP is essential for reducing oxidized glutathione by glutathione reductases, there is, unsurprisingly, a strong positive correlation between G6PD activity and glutathione and glutathione-S-transferase (GST) levels in cord blood [97].

Band 3 (anion exchanger 1; AE1; SLC4A1; capnophorin) is the most abundant RBC membrane protein. It is a chloride/bicarbonate anion transporter that also regulates RBC intermediary metabolism to allow adaptation to hypoxia and oxidant stress. Serving as a molecular “railway switch,” band 3 diverts glucose down different “tracks” (i.e., glycolysis vs. the PPP) depending on cellular needs. Deoxy-HbA competes for binding to the N-terminal cytosolic domain of band 3, thereby blocking the docking site for glycolytic enzymes, displacing them from the membrane, and boosting glycolysis. In contrast, at high oxygen saturations, glycolytic enzymes bind to the band 3 N-terminal, which favors glucose oxidation via the PPP to produce NADPH [98].

Importantly, HbF interacts more weakly with band 3 than HbA due to differences in binding between HbF and the band 3 binding cleft [99]. Given the pivotal role of band 3 in regulating RBC metabolism in vivo and during refrigerated storage, this could explain differences in fetal RBC adaptations to different oxygen concentrations and in regulating flux through glycolysis vs. the PPP. Sphingosine 1-phosphate (S1P) also regulates RBC metabolism during hypoxia. S1P binds and stabilizes deoxy-Hb, promoting release of glycolytic enzymes from band 3 into the cytosol, thereby promoting glycolysis. S1P serum levels are higher in cord blood as compared with adult blood and may thereby help modulate glycolysis and the PPP in fetal RBCs. [100].

## 7. Oxidative Injury to Neonatal RBCs During the Perinatal Period

During their lifespan, neonatal RBCs face the demanding challenge of surviving in a hostile environment while continuing to deliver oxygen efficiently; thus, they are simultaneously targets of extracellular oxidative stress and generators of ROS [101]. Therefore, neonatal RBCs contain multiple antioxidants to protect them and surrounding tissues from attack by ROS resulting from hemoglobin auto-oxidation [102] or external sources [103]. For example, RBC antioxidant enzymes (e.g., superoxide dismutase [SOD], catalase [CAT], glutathione peroxidase [GPX], and glutathione-S-transferase [GST]) and nonenzymatic antioxidants (e.g., glutathione [GSH], thioredoxins, vitamin C, vitamin E, and protein sulfhydryl groups) protect RBCs from ROS [104] (Figure 1(9)).

Overall, preterm and term neonatal RBCs possess abundant enzymatic antioxidant defenses [105]. CAT and GPX levels tend to increase with gestational age, whereas GST decreases [106]. CAT, GPX, and SOD levels at birth are slightly lower in preterm neonates, especially if delivered by caesarean section [107]. In addition, G6PD, glutathione reductase (GR), GPX, CAT, and SOD activities are comparable in term and preterm newborn RBCs at birth; G6PD and GR activities are even higher in term and preterm newborns than in adults, whereas GPX, CAT, and SOD activities are similar to those of adults [108]. GSH and GSSG concentrations are significantly higher in preterm and term newborn RBCs than in adults, but with a lower GSH/GSSG ratio, possibly reflecting increased exposure to oxidative stress beginning with delivery [109]. Supporting this hypothesis, CB-RBCs from both preterm and term neonates demonstrated more effective glutathione recycling after exposure to oxidant stress as compared with adult RBCs [109]. However, RBC GST levels rapidly decline in the first 3 h after birth in both term and preterm neonates [110]. Of note, in GSH recycling experiments, term and preterm newborn RBCs have a higher rate of GSH regeneration than adult RBCs when exposed to an oxidant challenge [108,111]. Consequently, fetal RBCs, due to their increased activity of GST, CAT, and SOD as compared with adult RBCs, provide better protection for cultured lymphocytes against chromosome breaks induced by diepoxybutane, [112]. Thus, the efficacy of GSH recycling in RBCs at birth protects tissues in premature babies from peroxidative damage [113] and/or compensates for tissue-deficient antioxidant capacity.

Cysteine, a key GSH precursor, may be a conditionally essential amino acid for preterm infants since cystathionine gamma-lyase, which catalyzes the last step of the trans-sulphuration pathway by cleaving cysteine from cystathionine, is undetectable in fetal liver, only appearing postnatally [114]. For example, GSH levels are lower in very low birth weight neonates as compared with controls, resulting from insufficient cysteine from maternofetal transfer [114]. Therefore, parenteral cysteine supplementation in preterm neonates is hypothesized to increase RBC GSH stores and availability, thereby decreasing markers of oxidative injury and inflammation and neonatal disease burden; however, results to date are limited and inconsistent [115,116].

Despite robust antioxidant defenses, RBCs in the perinatal period are particularly vulnerable to ROS that can irreversibly damage RBC proteins, lipids, and membranes, progressively altering RBC properties and hampering their function (Figure 1(10)) [103]. In several studies, ROS production by neonatal RBCs was increased as compared with adult RBCs when exposed to pro-oxidants, such as phenylhydrazine [117] and α-naphthol [94]. Key mediators of oxidative stress in the perinatal period include non-transferrin-bound iron (NTBI) and free iron in the RBC cytosol [101]. For example, free iron in the RBC cytosol inversely correlates with gestational age (i.e., higher in preterm newborns) and directly correlates with the degree of hypoxia to which RBCs were exposed [118]. After reoxygenation, hypoxic neonatal RBCs release large amounts of free iron, contributing to the appearance of NTBI in newborn plasma and increasing systemic oxidative stress [119]. Interestingly, exposing neonatal RBCs to hypoxia/reoxygenation-related injury elicits post-translational protein modifications aimed at protecting cellular components [120]. Specifically, several cytosolic proteins in newborn RBCs are Tyr-phosphorylated, including CAT, antioxidant protein 2 (AOP2), biliverdin IX-β reductase (BVR), and thioredoxin peroxidase 1 (Tpx 1), all involved in antioxidant defense. Others include actin and glyceraldehyde 3-phosphate dehydrogenase (G3PDH); indeed, phosphorylated G3PDH, released from band 3, may enhance glycolysis.

Neonatal RBC membranes seem particularly vulnerable to ROS-induced damage (Figure 1(10)). Indeed, neonatal RBCs show high levels of malondialdehyde adducts with phospholipids, and nearly twice the adult level of thiobarbituric acid reactivity, an index of lipid peroxidation, suggesting significant peroxidative membrane lipid damage in vivo [121]. Neonatal RBC membrane vulnerability can be due, in part, to its unique composition (i.e., rich in PUFAs with bis-allylic hydrogens, the preferred target of ROS [122]) and to the relative deficiency of membrane antioxidants such as vitamin E.

Membrane oxidation of RBCs from premature and term neonates is associated with several comorbidities. For example, elevated hydroperoxide concentrations were found in RBC membranes early in life, along with low levels of antioxidant defense mechanisms, with particularly low vitamin E levels in premature infants [123]. Vitamin E is the most important inhibitor of membrane lipid peroxidation [124], and levels in preterm infants at birth are generally comparable to, or at least not lower than, those in term infants; however, they quickly develop progressive deficiency due to low oral intake from feeding intolerance and increased exposure to oxidative stress; therefore, vitamin E supplementation was proposed to reduce the severity of comorbidities of prematurity [20].

Neonatal RBC membranes also contain higher membrane-bound hemoglobin (mb-Hb) levels than adult RBC membranes. The amount of mb-Hb increases when adult RBCs are incubated with increasing concentrations of hydrogen peroxide; however, neonatal RBC membrane proteins are more resistant to hydrogen peroxide. This may be because HbF adds to the reduction potential of neonatal RBCs, partially protecting their membrane proteins against oxidant stress [125]. Therefore, mb-Hb was proposed as a marker for ROS-induced injury, and its accumulation is associated with the aging process of neonatal RBCs [126]. To support this hypothesis further, oxidative stress biomarkers in urine were significantly higher in neonates with lower HbF levels due to RBC transfusions from adult donors, supporting the concept that HbF may help prevent free radical-associated pathology during the newborn period [127].

Finally, oxidative stress leads to band 3 oxidation and membrane clustering, thereby forming a neoantigen that binds autologous antibody and complement, allowing direct recognition by phagocytes [128]. These clusters occur much faster with newborn (particularly preterm) RBCs as compared with adult RBCs, especially in hypoxia, thereby accelerating clearance in vivo [129].

## 8. Exposome

Fetal and neonatal RBC physiology is profoundly affected by environmental factors related to pregnancy, the so-called “maternal exposome”. For example, although many compounds in cigarette smoke may be harmful to human health, 20–30% of women continue smoking during pregnancy. RBCs from neonates whose mothers smoked have higher levels of toxic peroxynitrite (ONOO−) generated by the reaction of superoxide anion (O2●−) and NO, and increased levels of lipid peroxidation. Chronic exposure to pro-oxidants derived from tobacco smoke affects the neonatal RBC lipidome, thereby impairing RBC deformability and rheology [130]. Moreover, these neonatal RBCs had impaired eNOS activity and increased arginase 1 levels [131], which alter crosstalk between RBCs and the endothelium, reducing NO release and ROS scavenging [132].

Gestational hypertension and diabetes, common complications of pregnancy, affect fetal RBCs. Thus, neonatal RBCs from mothers with pre-eclampsia have a lower elongation index, reflecting impaired deformability and lower aggregability; the latter may be an adaptation to endothelial dysfunction and increased vascular resistance, thereby maintaining blood flow [133]. Indeed, a prospective case-control study demonstrated that neonates born to mothers with early-onset pre-eclampsia had elevated blood pressures in the first month of life, compared with neonates born to mothers with normal blood pressure [134]. Similarly, in pregnant women with pre-existing or gestational diabetes, there are increased markers of oxidative stress (e.g., NO degradation products and total glutathione [135]) and decreased fetal levels of ARA, DHA, and other PUFAs [136]. The rheological properties of these fetal RBCs are altered, as compared with the non-diabetic setting, perhaps from nonenzymatic glycation of RBC proteins and oxidative membrane damage [137].

Fetal exposure to heavy metals from maternal circulation (e.g., mercury, lead, and cadmium [138]) may affect fetal and neonatal health in general, and fetal RBCs in particular. In addition, fetuses can be exposed to mercury and lead by intrauterine blood transfusions [139]; similar results also occur in transfused neonates [140]. Lead alters fetal RBC morphology, modifies energy metabolism, and increases oxidative stress by inhibiting glycolysis and the PPP [141]. Decreased purine nucleotides, ATP in particular, and increases in their catabolic products may provide metabolic markers of lead toxicity [141]. In contrast, cadmium induces the accumulation of 4-hydroxynonenal, a marker of RBC membrane damage [132].

Finally, when mothers reside in areas that expose them to high ambient levels of pesticides during pregnancy, fetal RBCs exhibit increased osmotic fragility and reduced SOD levels [142]. Finally, comprehensive studies of the effects of the exposome on the storage and transfusion quality of donated adult RBCs and CB-RBCs could improve the selection of units for fetal and neonatal transfusions.

## 9. How Neonatal RBCs Age and Die

The exact mechanisms determining RBC life span and regulating senescent RBC clearance are not completely understood [143]; this is particularly true in the perinatal period (Figure 2). Non-steady-state hematopoiesis, rapid infant growth, repeated phlebotomy, and multiple transfusions make RBC survival studies difficult to conduct and interpret in these vulnerable patients [144]. It is also plausible that RBC lifespan is not a fixed, intrinsic feature of RBCs but rather results from the interaction of biological properties with environmental factors, providing a flexible system to adapt RBC number to tissue oxygen demand.

Fetal hematopoiesis resembles stress hematopoiesis, which occurs in adults in response to acute hemolysis, hemorrhage, and hypoxia [145]; thus, there is a need for rapid production of new RBCs to meet metabolic demands of developing tissues. Similar to fetal RBCs, stress RBCs are larger and express more HbF. When the stress ends, the RBCs that formed during the stress event are selectively removed by neocytolysis [146]. The abrupt exposure to the extrauterine environment at birth may itself activate neocytolysis, contributing to decreases in Hb postnatally [147,148].

The survival of fetal RBCs in vivo has been investigated by various methods, including chromium-51 labeling. The median RBC life span in newborns was initially shown to be 80 days (compared to ~120 days in adults) [149]; more complex models and labeling techniques lowered this value to ~55 days [150,151]. The importance of the circulatory environment in modulating neonatal RBC lifespan was underscored by finding that fetal RBCs, from infants delivered at term after massive antenatal feto-maternal hemorrhage, can survive approximately twice as long in the maternal circulation as in the newborn infant [152]. In contrast, the mean lifespan of adult RBCs, after transfusion into neonates, was only 56.4 days [153]. In more recent studies using biotinylated adult RBCs, the mean lifespan was 85 days, which is only modestly shorter than that for stored human RBCs transfused into adult recipients (i.e., 103–116 days) [154,155]. Of note, in contrast to adult recipients, when adult RBCs were transfused into premature infants there were no decreases in 24-h post-transfusion RBC recovery as storage time increased [156]. When autologous neonatal RBCs and allogeneic adult RBCs were concurrently transfused to the same infant, the mean lifespan of the adult RBCs was 70.1 days, whereas it was 54.2 days for the neonatal RBCs [144]. Although differences between these RBC populations probably contribute to determining their lifespan, environmental factors may also be relevant. For example, the total number of complete circulatory circuits completed during the lifespan of an RBC (denoted as N-max) is relatively constant for all age groups, because every cycle of oxygenation/deoxygenation and deformation that RBCs experience contributes to membrane stress and senescence. Because circulation speed in neonates is faster than in adults, RBCs make more trips per unit of time and reach the N-max more quickly, thereby resulting in a shorter lifespan [151]. Nonetheless, additional RBC survival and lifespan studies are needed to validate these findings, including evaluating adult RBC units transfused into preterm neonates.

During aging, neonatal RBCs undergo extensive shape remodeling, with altered membrane composition and enzymatic activities, along with increasing rigidity and osmotic fragility [157,158]. Aged neonatal RBCs expose PS on their membrane along with progressive loss of sialic acid [159,160], and ATP-depleted senescent RBCs have increased cell surface immunoglobulin and complement, all of which enhance macrophage clearance [161]. However, decreased functioning of the reticuloendothelial system (in the spleen and elsewhere) was proposed to explain the increased numbers of pitted cells (“pocked RBCs”) seen in the neonatal period, as these abnormal RBCs are typically increased in splenectomized patients [162]. Indeed, the decreased ability of neonatal macrophages to remove transfused storage-damaged adult RBCs could explain the aforementioned absence of reduced 24-h post-transfusion RBC recovery in neonates as storage time increased [156]. An extreme consequence of accumulated sublethal damage to RBCs is the activation of a specific form of programmed cell death: eryptosis [163]. Eryptosis involves dysfunctional ion exchange, cell shrinkage, ceramide accumulation, cell membrane vesiculation due to cytosolic calcium overload, and membrane phospholipid scrambling with PS exposure [164]. Among factors inducing eryptosis, neonatal RBCs are more resistant to chloride removal, osmotic shock, prostaglandin E2, and platelet-activating factor, but are more sensitive to oxidative stress [165]. Taken together, these findings suggest that circulating senescent RBCs are, simultaneously, targets and generators of extracellular free radicals. For example, hemolysis of neonatal RBCs due to oxidative damage was implicated in the pathogenesis of neonatal jaundice of unknown etiology [166]. Notably, the levels of the natural heme scavengers haptoglobin and hemopexin tend to increase with gestational age [167], making preterm neonates more vulnerable to intravascular hemolysis-induced damage [168].

## 10. Allogenic CB-RBC Transfusions in Extreme Preterm Neonates

CB-RBCs have been increasingly used for pediatric and neonatal transfusions for both autologous and allogeneic purposes [169]. Allogeneic CB-RBC concentrates, obtained from cord blood units from full-term neonates donated to public blood banks, are particularly promising for meeting the transfusion demands of extreme preterm neonates (i.e., born before week 28 of gestation) who require transfusion support for prolonged periods. Repeated transfusions of standard RBC concentrates, containing mainly HbA, are associated with morbidity and mortality in preterm neonates [170]. One possible mechanism involves the progressive depletion of HbF and replacement with HbA from repeated transfusions of adult RBCs [171], given that HbF could be protective in preventing development of various comorbidities of prematurity, such as ROP and BPD [76,172,173,174]. In particular, in preterm neonates, the increased oxygen delivery from high levels of HbA (which offloads oxygen better than HbF) may expose neonatal tissues to hyperoxia, thereby producing excessive levels of ROS in the absence of adequate levels of antioxidants [175]. Therefore, transfusing allogenic CB-RBCs may be better for correcting anemia while maintaining physiological HbF levels [176]. Cerebral tissue oxygenation after transfusing standard or CB-RBC concentrates follows different patterns in preterm neonates, compatible with a hyperoxic condition in the case of adult-donor transfusions [177]. In addition, CB-RBCs differ from adult RBCs in ways other than HbF expression, including increased deformability and NO production; therefore, they may function more optimally in the preterm neonatal circulation, providing physiological benefits. Nonetheless, randomized clinical trials are needed to assess whether CB-RBC transfusions are better; for example, the ongoing multicenter randomized BORN trial will determine whether CB-RBC transfusions can reduce ROP severity in extremely low birth weight neonates [178].

In parallel with the growing interest in using CB-RBCs for transfusion, there are increasing efforts to optimize protocols for producing CB-RBC concentrates and define optimal storage conditions. Thus, after collection, whole cord blood is usually stored at room temperature for variable time periods before further processing; this is dictated by the time of birth, transportation to the processing facility, and laboratory hours. In an extensive evaluation of CB-RBC quality after up to 65 h of room temperature storage, including measuring hemolysis rate, deformability, vesiculation, surface expression of PS and CD47, and methemoglobin, 2,3-DPG, and ATP levels, the only change identified was decreased RBC 2,3-DPG levels [179]. In addition, based on prior experience, whichever protocol is adopted, the collection and fractionation of whole cord blood is feasible and can produce CB-RBC concentrates allowing the same transfusion dose (i.e., 20 mL/kg for preterm neonates) as adult-donor RBCs, with similar Hgb content and residual white blood cells after leukofiltration [180,181,182,183,184,185,186,187]. Nevertheless, during refrigerated storage, RBCs undergo multiple metabolic and morphological changes (i.e., the “storage lesion”); these have been extensively characterized for adult-donor RBCs [188]. Several studies have similarly evaluated stored CB-RBCs; for example, refrigerated storage of CB-RBCs in SAGM preservative solution for 14 days led to increases in potassium and lactate levels, free Hb, and percent hemolysis, and decreases in pH and glucose [185]. In addition, after 35 days of storage, intracellular ATP decreased significantly [182]. The hemolysis rate seems to increase dramatically after the first 3 weeks of storage [183]. Overall, CB-RBCs seem to have a shorter shelf life as compared with standard adult RBC units [184]. If γ-irradiation is performed to prevent transfusion-associated graft-versus-host disease, additional injury to CB-RBCs occurs, particularly to the RBC membrane [189], reducing the blood bank shelf life to 14 days. In addition, CB-RBC units have higher rates of microbial contamination than adult-donor RBC units and must be cultured for bacteria and fungi before distribution for clinical use, thereby delaying release by 5–6 days. To this end, pathogen reduction technologies, recently introduced for adult-donor RBC units [190], may be useful in this setting to reduce the risk of microbial contamination and possibly prolong CB-RBC shelf life. Importantly, cord blood units are currently collected and stored by blood and tissue facilities as sources of hematopoietic stem cells; in view of wider transfusion use, regulations for collection would need to be modified, and quality standards must be defined for CB-RBC concentrate processing, storage, and issue. In the interim, promising preliminary results with CB-RBC transfusions not only demonstrate that HbF-enriched blood products are safe, but patients receiving CB-RBCs also have less severe bradycardia and pulmonary hypertension [191]. Moreover, elucidating the time course of functional, morphological, and biochemical changes in CB-RBCs throughout refrigerated storage will expand our knowledge about CB-RBC physiology as compared to adult RBCs. Additionally, optimization of processing protocols, storage additive solutions, and storage conditions must occur. Hypoxic storage of CB-RBCs [192], mimicking the physiological conditions experienced by fetal RBCs, may also be a fruitful area for research.

Finally, it should be remarked that the limited yield and the persistent synthesis of HbF are among the biggest challenges in generating functional RBCs from embryonic stem cells or human-induced pluripotent stem cells [193]. Conversely, low-volume HbF-enriched RBC products could be more easily obtained in vitro, providing a possible alternative source for the transfusion therapy of extremely preterm neonates.

## 11. Conclusions

There are many fundamental differences between fetal and adult RBCs, including cell size, metabolism, deformability, and lifespan. As HbF and HbA exhibit differences in ROS production and oxygen offloading, transfusions of HbF-containing RBCs to fetuses and neonates during their critical periods of development may produce fewer adverse outcomes in these vulnerable patients.

## Figures and Tables

**Figure 1 cells-13-01843-f001:**
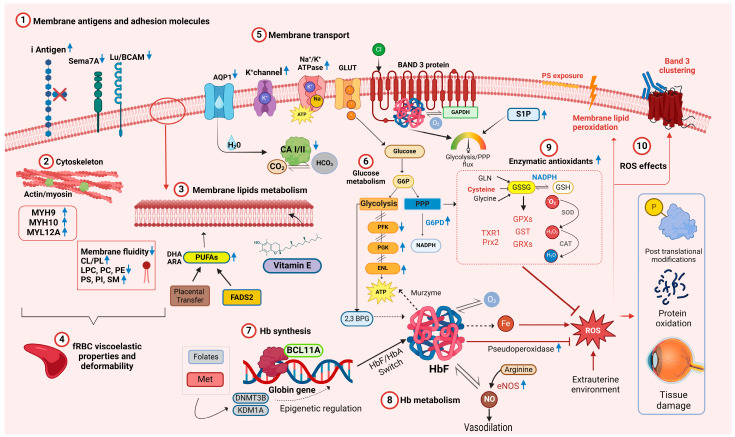
Overview of the main molecular networks and metabolic pathways relevant to fetal and neonatal RBC physiology. Blue arrows pointing upward denote increased concentration or activity in relation to adult RBCs. Blue arrows pointing downward indicate decreased concentration or activity. Abbreviations: AQP1 aquaporin 1, ARA arachidonic acid, ATP adenosine triphosphate, BCL11A B-cell lymphoma/leukemia 11A, CA I/II carbonic anhydrase I/II, CAT catalase, Cl− chloride ion, CL cholesterol, CO_2_ carbon dioxide, DHA docosahexaenoic acid, DNMT3B DNA methyltransferase 3B, ENL enolase, eNOS endothelial nitric oxide synthase, FASD2 fatty acid desaturase 2, fRBC fetal RBCs, GAPDH glyceraldehyde-3-phosphate dehydrogenase, GLN glutamine, GLUT glucose transporter, GSSG glutathione disulphide, GSH glutathione, GPX glutathione peroxidase, GST glutathione S-transferase, GRXs glutaredoxins, Hb hemoglobin, HbA adult hemoglobin, HbF fetal hemoglobin, HCO3 bicarbonate, H_2_O water, H_2_O_2_ hydrogen peroxide, LPC lysophosphatidylcholines, Lu/BCAM Lutheran blood group and basal cell adhesion molecule, K+ potassium ion, KDM1A lysis Met methionine, MYH9 myosin heavy chain 9, MYH10 myosin heavy chain 9, MYL12A myosin regulatory light chain 12A, Na+ sodium ion, NO nitric oxide, O_2_ molecular oxygen, •O_2_ superoxide, PC phosphatidylcholine, PE phosphatidylethanolamine, PFK phosphofructokinase-1, PGK phosphoglycerate kinase, PI phosphatidylinositol, PS phosphatidylserine, Prx2 Peroxiredoxin 2, ROS reactive oxygen species, Sema-7A Semaphorin 7A, SM sphingomyelin, SOD superoxide dismutase, TXR1 thioredoxin 1, 2,3-DPG 2,3-diphosphoglycerate. Created in BioRender.com.

**Figure 2 cells-13-01843-f002:**
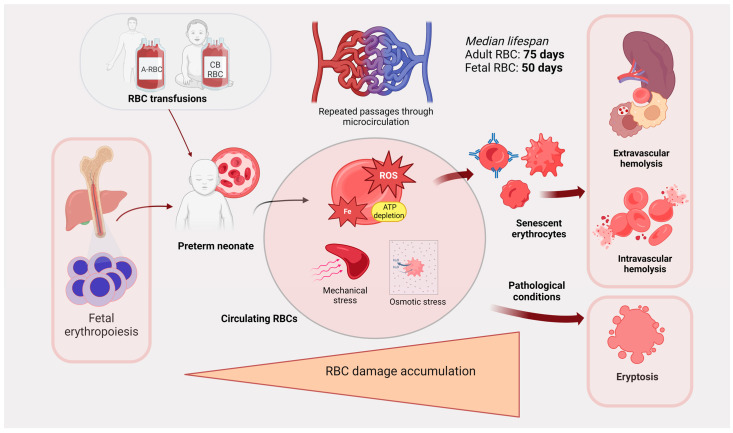
Schematic demonstrating how RBCs age and die in preterm neonates. RBCs from hematopoiesis or transfusion circulate through the microvasculature in preterm neonates. Considerable RBC damage accumulates over time from ROS, mechanical injury, and osmotic stress, and this leads to RBC senescence. RBCs are then cleared by extravascular or intravascular hemolysis and by eryptosis. ROS: reactive oxygen species. Created in BioRender.com.

**Table 1 cells-13-01843-t001:** Overview of hematological indices and morphological parameters of fetal RBCs compared to their adult counterparts.

Parameter	Findings	Reference
*Mean corpuscular volume (MCV)*	Increased	[1,2,3,11,13]
*Mean corpuscular hemoglobin (MCH)*	Increased	[1,2,3,13]
*Mean corpuscular hemoglobin content (MCHC)*	Comparable	[1,2,3]
*Red blood cell distribution width (RDW)*	Increased	[8]
*Reticulocyte count*	Increased	[9]
*Surface-area-to-volume ratio*	Decreased	[11,13]
*Sphericity*	Increased	[13]
*Membrane fluctuations*	Comparable	[13]
*RBCs with altered morphology (%)*	Increased	[10]

## Data Availability

No new data were created or analyzed in this study. Data sharing is not applicable to this article.

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
