# Peer review of "Fetal Red Blood Cells: A Comprehensive Review of Biological Properties and Implications for Neonatal Transfusion"

_cells, 2024, doi:10.3390/cells13221843_

Round 1

Reviewer 1 Report

Comments and Suggestions for Authors

The manuscript entitled "Fetal red blood cells: a comprehensive review of biological properties and implications for neonatal transfusion" by C. Pellegrino et al. in a comprehensive, and yet a concise way presents the up-to-date knowledge about various features of the erythrocytes in preterm neonates, in the context of blood transfusion.

The work is divided into 11 sections, each focusing on the differences between given adult and fetal/neonate RBC functionality. The manuscript containes two figures, one of them being a complicated overview of molecular networks and metabolic pathways relevant to fetal and neonate RBC physiology, and the second one depicting the process of neonatal RBC ageing and dying. 

Among 191 of the cited references, around 45% are the works published within the last ten years.

Although the text is clear and easy to understand, there are some minor language/editorial issues that should be corrected before publication:

-both in the Abstract and the Introduction it should be emphasized that the transfusion of, lets say, "adult" RBC in neonates impact the physiology of neonatal RBC.

-line 39- the first part of the sentence lacks the predicate, i.e. a verb  describing the process of creating the large laboratory datasets;

-line 62- as same as above, the lack of predicate describing that something was done with use of conventional light microscopy;

-line 73- a sentence should not  start with a preposition  such as 'by'. Instead, the sentence should start like :"Quantitative phase imaging of cord RBCs revealed, that the RBCs of... " and so on. " It is to stress out that something was done with the use of such and such method. 

-line 148- "i antigen" insted of "I antigen"- please correct

-line 366- are there any Suplementary Materials that come with the Review, or is it just an accidental insert?

-line 545 - an additional space in the brackets between references 160 and 161;

-line 607- an uncecessary space and hyphen- please remove.

Author Response

We thank the Reviewer for the constructive comments.

1) both in the Abstract and the Introduction it should be emphasized that the transfusion of, lets say, "adult" RBC in neonates impact the physiology of neonatal RBC.

As suggested, we emphasized that “transfusions of adult RBCs in neonates……deeply modify the physiology..”.

2) -line 39- the first part of the sentence lacks the predicate, i.e. a verb  describing the process of creating the large laboratory datasets;

- we modified the sentence as follows: “Modern hematology analyzers produce large laboratory datasets that allow to define of reference ranges for RBC indices in neonates born at different gestational ages”

3) -line 62- as same as above, the lack of predicate describing that something was done with use of conventional light microscopy;

We modified the sentence as follows: “Term and preterm newborn RBCs display at conventional light microscopy, more morphological anomalies”

 4) -line 73- a sentence should not  start with a preposition  such as 'by'. Instead, the sentence should start like :"Quantitative phase imaging of cord RBCs revealed, that the RBCs of... " and so on. " It is to stress out that something was done with the use of such and such method. 

We modified the sentence as suggested.

5) -line 148- "i antigen" insted of "I antigen"- please correct

i has been changed to I

6) -line 366- are there any Suplementary Materials that come with the Review, or is it just an accidental insert?

It was an accidental insert. We apologize for the inaccuracy.

7 and 8) -line 545 - an additional space in the brackets between references 160 and 161;

-line 607- an uncecessary space and hyphen- please remove.

Both typos have been corrected.

Reviewer 2 Report

Comments and Suggestions for Authors

This is an excellent review and covers an area that partly has been negelcted. It certainly deserves to be published. There are only some minor points that the authors may consider:

The manuscript does not provide any details on the composition of the neonate plasma. For instance, the levels of haptoglobin and hemopexin could be discussed because these proteins are deeply involved in the degradation process of Hb after hemolysis. This may in turn also influence potential inflammatory side reactions of the placenta.

It would also be valuable to include a more detailed overview of RBC development from stem cells and embryonic RBC to generate a better understanding of the cellular behaviour.

Besides these points the manuscript is fine, well written and includes relevant references.

Author Response

This is an excellent review and covers an area that partly has been negelcted. It certainly deserves to be published. There are only some minor points that the authors may consider:

1) The manuscript does not provide any details on the composition of the neonate plasma. For instance, the levels of haptoglobin and hemopexin could be discussed because these proteins are deeply involved in the degradation process of Hb after hemolysis. This may in turn also influence potential inflammatory side reactions of the placenta.

We thank the Reviewer for this comment. Accordingly, we added the following sentence “Notably, the levels of natural heme-scavengers haptoglobin and hemopexin tend to increase with gestational age [167], making preterm neonates more vulnerable to the intra-vascular hemolysis-induced damage [168].” (line 559). Two new references have been included in the review.

 2) It would also be valuable to include a more detailed overview of RBC development from stem cells and embryonic RBC to generate a better understanding of the cellular behaviour.

We thank the Reviewer for this thought-provoking suggestion. We added the following sentence at the end of the last paragraph: “Finally, it should be remarked that the limited yield and the persistent synthesis of HbF are among the biggest challenges in generating functional RBCs from embryonic stem cells or human induced pluripotent stem cells [194]. Conversely, low-volume HbF-enriched RBC products could be more easily obtained in vitro, providing a possible alternative source for the transfusion therapy of extremely preterm neonate". A new reference has been added (ref 194 Lee SJ, Jung C, Oh JE, Kim S, Lee S, Lee JY, Yoon YS. Generation of Red Blood Cells from Human Pluripotent Stem Cells-An Up-date. Cells. 2023 Jun 5;12(11):1554.)

Reviewer 3 Report

Comments and Suggestions for Authors

The review article authored by Pellegrino et al. titled “Fetal red blood cells: a comprehensive review of biological 2 properties and implications for neonatal transfusion” collates a wide range of biochemical and functional phenotypic properties of fetal/neonatal red blood cells (RBC). Furthermore, the authors have discussed the potential implications of replacing these red cells with adult red cells. The draft review article provides a comprehensive understanding of the molecular networks and metabolic pathways in fetal/neonatal RBCs. The manuscript is well-written and provides evidence-based insights into the peculiarities of fetal/neonatal RBCs relevant to transfusion medicine. The illustrations in the draft review nicely summarize the key pathways and pathophysiologic modulators of neonatal/fetal RBCs. I have a few minor comments for the authors to consider:

-            The terms fetal and neonatal seem to be used interchangeably in some parts of the manuscript e.g. lines 557-559.

-            It is recommended that the laboratory indices discussed in Section 2 (Fetal RBC shape, hemoglobin content, and morphology) be listed in a table with references to make it easier for readers.

-            Figure 2 seems to be a little confusing since mechanisms of physiological senescence seem to overlap with eryptosis. However, there is not much evidence to support the latter process since it is primarily believed to be involved in pathologic states and not per se a regulator of RBC lifespan in healthy states. It is true that multifactorial mechanisms of RBC clearance have been posited so the distinction between senescence and eryptosis is relatively vague.

Author Response

The review article authored by Pellegrino et al. titled “Fetal red blood cells: a comprehensive review of biological 2 properties and implications for neonatal transfusion” collates a wide range of biochemical and functional phenotypic properties of fetal/neonatal red blood cells (RBC). Furthermore, the authors have discussed the potential implications of replacing these red cells with adult red cells. The draft review article provides a comprehensive understanding of the molecular networks and metabolic pathways in fetal/neonatal RBCs. The manuscript is well-written and provides evidence-based insights into the peculiarities of fetal/neonatal RBCs relevant to transfusion medicine. The illustrations in the draft review nicely summarize the key pathways and pathophysiologic modulators of neonatal/fetal RBCs. I have a few minor comments for the authors to consider:

1) The terms fetal and neonatal seem to be used interchangeably in some parts of the manuscript e.g. lines 557-559.

We thank the Reviewer for this comment: accordingly, we modified "fetal" into "neonatal", which was more appropriate according to the cited reference.

2) It is recommended that the laboratory indices discussed in Section 2 (Fetal RBC shape, hemoglobin content, and morphology) be listed in a table with references to make it easier for readers.

We thank the Reviewer for this constructive comment. We added the Table 1, as suggested.

3)  Figure 2 seems to be a little confusing since mechanisms of physiological senescence seem to overlap with eryptosis. However, there is not much evidence to support the latter process since it is primarily believed to be involved in pathologic states and not per se a regulator of RBC lifespan in healthy states. It is true that multifactorial mechanisms of RBC clearance have been posited so the distinction between senescence and eryptosis is relatively vague.

We agree with the Reviewer and modified Figure 2 to emphasize that eryptosis is primarily involved in pathologic states.